# Uncertainty Analysis of a Test Bed for Calibrating Voltage Transformers Vs. Temperature [note 1]

**DOI:** 10.3390/s19204472

**Published:** 2019-10-15

**Authors:** Alessandro Mingotti, Lorenzo Peretto, Roberto Tinarelli, Abbas Ghaderi

**Affiliations:** Department of Electrical, Electronic and Information Engineering, Alma Mater Studiorum—University of Bologna, 40126 Bologna BO, Italy; lorenzo.peretto@unibo.it (L.P.); roberto.tinarelli3@unibo.it (R.T.); abbas.ghaderi2@unibo.it (A.G.)

**Keywords:** accuracy, accuracy class, instrument transformer, metrological characterization, phase error, ratio error, temperature, test bed

## Abstract

The paper addresses the evaluation of the uncertainty sources of a test bed system for calibrating voltage transformers vs. temperature. In particular, the Monte Carlo method has been applied in order to evaluate the effects of the uncertainty sources in two different conditions: by using the nominal accuracy specifications of the elements which compose the setup, or by exploiting the results of their metrological characterization. In addition, the influence of random effects on the system accuracy has been quantified and evaluated. From the results, it emerges that the choice of the uncertainty evaluation method affects the overall study. As a matter of fact, the use of a metrological characterization or of accuracy specifications provided by the manufacturers provides respectively an accuracy of 0.1 and 0.5 for the overall measurement setup.

## 1. Introduction

As it is well-known, a key element of distribution networks for measurement and protection purposes is the Instrument Transformer (IT), which was regulated by the Standard series from the International Electrotechnical Commission (IEC) IEC60044. Nowadays, after the introduction of a new kind of transformer, the so-called Low-Power Instrument Transformers (LPITs), most of the Standards have been replaced by the IEC 61869 series. In particular, IEC 61869-2 to 4 [1,2,3], still refer to the inductive voltage and current transformers (VT and CT), while IEC 61869 from 5 to 15 deal with all the unconventional voltage and current transformers (Rogowski coils, capacitive or resistive dividers, etc.) typically referred to as sensors. 

The introduction of LPITs changed the way to approach ITs outputs; as a matter of fact, they consist of few milli-ampere or few volts (hence loads in the order of fraction of VA) which means that they can be directly connected to the typical acquisition boards on the market. On the contrary, traditional inductive ITs need a further element to adapt their output (higher than some VA) to the acquisition system.

In addition to the aforementioned, this new kind of transformer has other interesting features compared to the old generation. Higher bandwidth limits, for example, is a feature that answers to new requirements coming from the spread of Distributed Generation (DG) along the power network. Due to DG presence, voltages and currents may contain harmonics up to 100 kHz, hence the measurement equipment must also behave correctly in that range of frequencies [4,5].

Approaching the ITs from a protection point of view, the inductive ones are not suitable for being connected to fast tripping switches or for recording transients in an accurate way [6,7,8,9,10].

Despite all the benefits introduced by LPITs, there is still a critical application for which utilities do not rely on them: metering for pricing. Legal metrology in energy and power measurements still use inductive ITs due to their robustness and stability, especially in terms of immunity vs. external electric and magnetic fields. However, even if relevant IEC Standards require several tests to be passed before the in-field implementation (e.g. accuracy vs. frequency), no verification of ITs accuracy vs. temperature is prescribed. Therefore, it is clear that metering for pricing may be significantly affected by an increase of ITs uncertainty caused by temperature variation. Hence, overall, there are still open topics for continuing the research on traditional inductive transformers.

Typically, ITs’ accuracy is not affected by the working temperature, however in some cases it could happen that the ITs’ accuracy parameters overcome the threshold limits.

In light of this, Mingotti et al. started dealing with the topic in [11,12] where setups for testing inductive current and voltage transformers vs. temperature have been developed. Moreover, in literature the metrological performances of transformers vs. temperature topic is tackled only for power transformers [13,14] but not for the instrument ones. In addition, to the authors’ best knowledge, no studies that focused on the accuracy vs. temperature behaviour of ITs are available.

In this paper, the analysis of the uncertainty sources which characterize the calibration setup presented in [11], is developed starting from the preliminary results obtained in [12]. In literature, different methods for approaching the uncertainty evaluation can be found [15,16,17]. Moreover, the investigation on the sources of uncertainty is a typical task when the accuracy evaluation is concerned, e.g. in [18] such evaluation has been performed on a three-phase state estimator for distribution networks.

Starting from the experience and knowledge gained in a previous work [19], the authors tackled the uncertainty issue following two distinct paths. Firstly, all the items composing the measurement setup have been metrologically characterized and the results have been used to compute the uncertainty affecting the ratio error and the phase displacement of the voltage transformer under test. Secondly, the same analysis has been carried out starting from the nominal accuracy specifications of each item. Both of them have been performed by applying the well-known Monte Carlo (MC) method, as is done in the majority of the uncertainty analysis [20]. Then, the two methods, which apply to the same measurement setup, are compared to understand whether they provide the same results or not. Such comparison is performed after the evaluation of random effects affecting the measurement setup.

The paper structure consists of: Section 2, the calibration system is recalled. Section 3 presents the uncertainty sources analysis. In particular, subsection A deals with the one obtained from the nominal specifications, while subsection B with the uncertainty obtained from the metrological characterization of the devices of which the above system is composed. In Section 4, uncertainty evaluation, using both the aforementioned methods, on ratio error and phase displacement is presented. Finally, Section 5 contains conclusion and some comments arising from the results of this work.

## 2. Measurement System

Two VTs, from here on referred to as A and B for the sake of manufacturers’ privacy, have been evaluated in terms of accuracy by using the test setup developed in [11] and shown in Figure 1.

Both VTs have a ratio of 20,000/100 ($K_{t}=200$) with variable accuracy (0.2 or 0.5) depending on the selected burden (10 and 25/30 VA, for transformer A and B, respectively). For our purposes, the former burden has been selected, resulting in a resistance of 330 Ω.

The experimental setup consists in the following main elements:A programmable calibrator Fluke 6105 A, which features up to 1000 V RMS, 20 A with an accuracy of 50 ppm for the amplitude and 10 µrad for the phase, from DC to 1 kHz. It provides sinusoidal stable inputs in terms of both amplitude and frequency to the transformer under test (TUT);A 0.1/15 kV VT used to increase the output voltage of the calibrator. This VT is used to adapt the Fluke output to the TUT rated voltage;The TUTs; both of the used VTs have 10 kV nominal voltage;A temperature-controlled chamber, where the VTs under test are placed for being tested in a temperature range between +5 °C and +60 °C;An 11:1 ratio resistive voltage divider. It consists of resistors featuring a thermal drift below 5 ppm/°C and relative accuracy of ± 10^−4^. The resistive divider is composed by the series of two 100 kΩ resistors at the high voltage terminals and by the series of two 10 kΩ resistors at the low voltage ones. This turns into a rated accuracy of the resistive divider equal to 0.02%. The aim of the divider is to adapt the TUTs’ output, which is $100/\sqrt{3}$ at rated voltage, to a value consistent with the input range of the Data AcQuisition board (DAQ), which is ±10 V_pp_;A 0.1 accuracy class, 5981:1 reference divider (R-C divider).A 24-bit DAQ NI9239 from National Instruments (Austin, USA); it acquires both the TUT and the dividers outputs;A laptop, to collect the data digitized by the DAQ and, by using the LabView software, to compute the desired parameters.

In brief, the series of the Fluke and the step-up transformer are used to provide the rated voltage to the TUTs. Then, two voltage dividers scale down the primary and secondary voltages of the TUTs before being acquired by the DAQ and elaborated with the laptop.

## 3. Uncertainty Sources Analysis

In this Section, two different analyses of the uncertainty sources are described and compared. The first one starts from the accuracy specification provided by the manufacturers of each component of the calibration system. The second one exploits their metrological characterization.

### 3.1. Nominal Accuracy Specification

The calibration system showed in Figure 1, and described in Section 2, is used to compute the ratio and phase errors of the TUT by comparing its performance with the ones of a voltage reference. Each element of the system introduces one or more sources of uncertainty which propagate along the measurement chain. The main sources of uncertainty are originated by the R-C voltage divider, the resistive divider and the acquisition board. The nominal accuracy specifications of such elements are summarized in Table 1.

### 3.2. Metrological Characterization

As shown in Figure 1, the voltage waveforms sampled by two channels of the DAQ are processed to get ratio error and the phase displacement of the TUT. Therefore, the following procedure has been adopted. The two sub-measurement systems consisting in R-C divider + DAQ and R-divider + DAQ have been singularly characterized.

In Figure 2 the calibration setup for the former sub-system is presented: The Fluke calibrator feeds the divider and its output waveform is collected by the DAQ and stored in a laptop, omitted in the picture for the sake of brevity. In total, 4 RMS voltages (1000, 650, 300, 100 V) have been tested.

The maximum voltage applied during the test was 1000 V rms. This is due to the output limits of the calibrator. However, from the divider calibration certificate a high linearity up to 20 kV can be assumed: the non-linearity error is not greater than 1/10,000.

For five days, 100 measurements were performed each day to guarantee the test repeatability. The calibrator has been used as a reference for both the amplitude and the phase of the voltage applied to the divider. Then, its actual ratio $k_{RC}$ and phase displacement $\varphi_{RC}$ have been calculated:(1)$$k_{RC}=\frac{\left|{\bar{V}}_{cal}\right|}{\left|{\bar{V}}_{RC}\right|},$$
(2)$$\varphi_{RC}=\hat{{\bar{V}}_{RC}}-\hat{{\bar{V}}_{cal}},$$
where $\left|{\bar{V}}_{RC}\right|$ and $\left|{\bar{V}}_{cal}\right|$ are the amplitudes of the output voltages of the R-C divider and the calibrator, respectively. $\hat{{\bar{V}}_{RC}}$ and $\hat{{\bar{V}}_{cal}}$ refer to the associated voltage phases. The relevant results are shown in Table 2 only for the case of 1000 V input voltage, for the sake of brevity. In the Table are listed, for each day, the actual mean ratio ${\tilde{k}}_{RC}$ of $k_{RC}$ and the mean phase displacement ${\tilde{\varphi }}_{RC}$ of $\varphi_{RC}$ introduced by the divider and the DAQ, together with their standard deviations.

From Table 2, it can be observed that even if the standard deviation obtained in the daily measurements are very low, daily changes of about 0.004% of ${\tilde{k}}_{RC}$ happen. This may lead us to conclude that, during the measurement procedure, some influence quantities have minor daily variations (i.e., room temperature). Therefore, the R-C divider has been characterized by the mean value and the standard deviation of the whole set of 500 measurements, which are 5983.27 and 0.07, respectively. Conversely, the day-to-day change of ${\tilde{\varphi }}_{RC}$ are consistent with the relevant standard deviations. However, mean value and standard deviation evaluated over the entire set of measurements has been considered: 0.22 mrad and 0.08 mrad, respectively.

Performance of R-divider + DAQ system have been verified in the second test. In Figure 3 the system is depicted; it consists of three components: the calibrator, the resistive divider and the DAQ. One hundred measurements have been repeated, each day, for 5 consecutive days. To approximate the nominal secondary voltage of the TUT ($100/\sqrt{3}$), 57 V RMS have been applied in each test. Following this, the ratio of the R-divider $k_{R}$ and the associate phase displacement $\varphi_{R}$ have been computed. With the same notation used for Table 2, Table 3 lists the aforementioned values and their standard deviations. From such results, it can be observed that daily variation of the mean value of both $k_{R}$ and $\varphi_{R}$ are within the intervals determined by the corresponding standard deviations $\sigma_{kR}$ and $\sigma_{\varphi R}$. Nonetheless, mean value and standard deviation computed over the entire set of measurements have been considered: 11.0024 and 2 × 10^−4^ for $k_{R}$ and −0.1 mrad and 0.2 mrad for $\varphi_{R}$.

To compute the aforementioned values’ uncertainty, both the experimental standard deviations and the calibrator uncertainty have to be considered. However, the latter contribution is at least one order of magnitude lower the former, hence it is negligible in the overall computation.

## 4. Uncertainty Evaluation

According to [2], the accuracy of a measuring VT is expressed by its accuracy class, which fixes maximum and minimum limits for the ratio (or voltage) error *ε* and the phase displacement Δ*φ*. From [2] it is:(3)$$\varepsilon =\frac{kU_{s}-U_{p}}{U_{p}},$$
(4)$$\Delta \varphi =\varphi_{s}-\varphi_{p},$$
where *U_p_* and *U_s_* are the primary and the secondary RMS voltages, respectively. *φ_p_* and *φ_s_* are the primary and the secondary phases of the primary and secondary phasors, respectively. Finally, *k* refers to the nominal ratio of the VT. As for *ε* and Δ*φ*, according to [2], their values should remain within the interval defined for each accuracy class. This holds for any voltage between 80% and 120% of the rated voltage and for burdens values between 0% and 100% of the rated one (and power factor = 1) in the case of burdens ≤10 VA.

The uncertainty on *ε* and Δ*φ* is estimated by starting from both the nominal accuracy specifications and the results of the metrological characterization, considered as uncertainty sources. The MC method has been used for the uncertainty evaluation of the cases studied in [11], which *ε* and Δ*φ* results are recalled, for the sake of clarity in Table 4. In particular, for all the listed cases, the uncertainty has been evaluated with the two approaches.

To completely understand the authors’ choice in the uncertainty evaluation process, it is worth mentioning that another approach that could have been applied to the presented measurement setup is the so-called “ratiometric” (or comparator) approach. It consists in the direct evaluation of ε and Δφ by using the ratio between the reference and the device under test quantities. In detail, as demonstrated in [21,22], the complex error variation Δε, defined as the difference between the complex error of the device under test ${\bar{\varepsilon }}_{x}$ and the complex error of the reference ${\bar{\varepsilon }}_{c}$, is:(5)$$\Delta \varepsilon ={\bar{\varepsilon }}_{x}-{\bar{\varepsilon }}_{c}=1-\frac{{\bar{V}}_{x}k_{x}}{{\bar{V}}_{c}k_{c}},$$
where ${\bar{V}}_{x}$ and ${\bar{V}}_{c}$ are the output quantities (phasors) measured from the device under test and the reference, respectively; whereas $k_{x}$ and $k_{c}$ are respectively the former and the latter ratios. In other words, $k_{x}$ and $k_{c}$, represent the known ratios of the devices used to perform the measurement of ${\bar{V}}_{x}$ and ${\bar{V}}_{c}$ (necessary for the Δε computation).

In light of Equation (5), when the uncertainty evaluation is concerned, two comments arise. Firstly, the computation complexity is higher, compared to the adopted approach, due to the presence of the phasors. Secondly, for $k_{x}$ and $k_{c}$ there are two options: the nominal values can be used, hence the uncertainty related to them is the nominal one, or a characterization of all the devices (as performed by authors in the previous section to obtain $k_{R}$ and $k_{RC}$) is necessary. Therefore, overall, adopting a ratiometric approach, in the authors’ opinion, would have not improved the quality of the results but only increased the computational effort.

### 4.1. Uncertainty Evaluation from Nominal Accuracy Specifications

As stated in Supplement 1 of the “Guide to the Expression of Uncertainty in Measurements”, [23,24], the application of the MC method requires the definition of a mathematical model of the measurement process. To this purpose, for this particular uncertainty evaluation, the model is as follows. The terms $\varphi_{s}$, $\varphi_{p}$, $U_{s}$, and $U_{p}$ have been obtained applying the Fast Fourier Transform to two different signals ${\bar{V}}_{s}$ and ${\bar{V}}_{p}$. These are two sinusoidal signals, to simulate the signals acquired by the DAQ, manipulated in order to include the appropriate sources of uncertainty:(6)$${\bar{V}}_{s}=({\bar{v}}_{s}+\alpha_{ns}+{\bar{v}}_{s}\alpha_{gs}+{\bar{v}}_{s}\alpha_{ds})K_{d}K_{t},$$
(7)$${\bar{V}}_{p}=({\bar{v}}_{p}\left(1+\alpha_{gp}+\alpha_{r}\right)+\alpha_{np})K_{r},$$
where ${\bar{v}}_{p}$ and ${\bar{v}}_{s}$ are the two sinusoidal signals; ${\bar{v}}_{p}$ already contains the shift angle due to the reference divider (element (6) in Figure 1). The α terms are the i-th values of uniformly distributed random variables; as specified in [24]. In particular, referring to Table 1, $\alpha_{ns}$ and $\alpha_{np}$ are the terms for the noise, $\alpha_{ds}$ refers to the divider contribution, $\alpha_{gp}$ and $\alpha_{gs}$ are the gain contributions from the DAQ. Two different values for the noise and the gain have been used in (6) and (7) because the adopted measured system collects the output voltages from two different channels.

In light of such uncertainty modelling procedure, 100000 MC trials have been performed. The results are listed in Table 5, where for both *ε* and Δ*φ*, the mean values, the standard deviation, the 95%-confidence interval and the uncertainty are shown, in column 2, 3, 4, and 5 respectively. 

Of course, the uncertainty is determined as half of the 95%-confidence interval width. Moreover, the probability distribution of *ε* and Δ*φ* are reported in Figure 4 and Figure 5, respectively. From the results, it can be stated that the proposed calibration setup, when not metrologically characterized, can be only used to evaluate *ε* and Δ*φ* of VTs belonging to an accuracy class not better than 0.5. As a matter of fact, the uncertainties on *ε* and Δ*φ* are about 1 $\times 10^{-3}$ and 1.45 mrad (95%-confidence interval), respectively. In light this, to consider an IT as belonging to a certain accuracy class, the uncertainty affecting the measurement of its *ε* and Δ*φ* must lay within 1/5 of the limits specified by the accuracy class itself. Hence, in light of the obtained results, the proposed setup can be adopted to test VTs up to the 0.5 accuracy class, which features limits of 0.5% and 6 mrad for *ε* and Δ*φ*, respectively. The above consideration is consistent with the use of a reference voltage divider with 0.1 accuracy class, as done in the proposed setup. This means that, to reduce the uncertainty in Table 6, a voltage divider with better accuracy class should be adopted as reference. Given that 0.1 is the best standardized accuracy class, the resulting solution is to move towards the metrological characterization of, at least, the voltage reference.

### 4.2. Uncertainty Evaluation from Metrological Characterization

By using the data obtained from the metrological characterization process presented in Section 2, 100,000 trials of the MC method have been run to compute the uncertainty of both ratio and phase errors. As it has been done in Section 4.1, the uncertainty modelling procedure must be described. However, the computation, starting from the results of a metrological characterization, is much simpler than the previous case. In fact, Equations (3) and (4) can be rewritten as:(8)$$\varepsilon =\frac{K_{t}\alpha_{kd}u_{s}-\alpha_{kr}u_{p}}{\alpha_{kr}u_{p}},$$
(9)$$\Delta \varphi =\varphi_{s}+\varphi_{pd}+\alpha_{pd}-\left(\varphi_{p}+\varphi_{pr}+\alpha_{pr}\right),$$
where $u_{s}$ and $u_{p}$ are the rms values of two voltages generated from the *ε* and Δ*φ* listed in Table 4. As for $\alpha_{kd}$ and $\alpha_{kr}$, they are the i-th values of uniformly distributed random variable obtained from the ratio of the resistive and reference divider, respectively. In Equation (4) instead, the phase shifts introduced by the resistive divider and the reference are $\varphi_{pd}$ and $\varphi_{pr}$; while their source of uncertainty are again random variables, uniformly distributed, referred to as $\alpha_{pd}$ and $\alpha_{pr}$.

Adopting the notation used in subsection A, and starting from the data listed in Table 4, results are shown in Table 6. It contains *ε* and Δ*φ* for the three temperatures and the two TUT used in [11]. As for Figure 6 and Figure 7, they represent, for the TUT A and temperature 25 °C, the estimated PDF for *ε* and Δ*φ*, respectively. The uncertainty, reported in the fourth column of Table 6, affecting the measured parameters is about one order of magnitude lower than that obtained when the nominal accuracy specifications are considered. Consequentially, we may assume that *ε* and Δ*φ* of VTs up to 0.1 accuracy class can be obtained by using the developed calibration setup. Such an assumption holds if the random variation of the *ε* and Δ*φ*, during their measurements, is comparable with the uncertainties due to the calibration system.

### 4.3. Evaluation of Random Effects

The random variation of *ε* and Δ*φ* has been already carried out in [9], where such quantities have been evaluated by performing 100 measurements in different working conditions. For the case analysed in this paper, the results are shown in Table 4 and the standard deviation represents the quantification of the considered effect. Given that, according to Figure 8, the distribution of the measured ratio error is approximatively gaussian, the contribution to uncertainty (95%-coverage probability) due to the random effects can be assumed equal to 2.7 × 10^−7^ = 1.4 × 10^−6^. The same considerations hold for the measured phase error, whose PDF is not shown for the sake of brevity. In such a case the contribution to uncertainty (95%-coverage probability) due to random effects can be assumed equal to 2.6 × 10^−4^ = 1.2 × 10^−3^ mrad.

In both cases, it can be stated that the contribution of the random effects is negligible compared to the calibration system uncertainties and hence, it is confirmed that VTs with an accuracy class of 0.1 or higher can be calibrated.

## 5. Conclusions

By considering the still intense usage of inductive instrument transformers, in particular when the measurement is used for legal metrology or billing purposes, authors focused on a specific feature of them: the accuracy vs temperature. Firstly, a calibration setup presented in a previous paper has been recalled. In such a paper, it has been verified that the calibration system allows the appreciation of a very low variation of the ratio error and phase displacement. As a matter of fact, the relevant standard deviations, representing the effect of random sources of uncertainty, are several orders of magnitude lower than the measured quantities. In this paper, starting from the conference paper, the study on the measurement setup uncertainty sources has been extended and concluded. This has been performed by by exploiting two approaches for the uncertainty evaluation. In particular, the comparison has been performed between the uncertainty results obtained from the metrologically characterization of the system and from the results obtained starting from the nominal accuracy specification provided by the manufacturers of the devices. From the metrological characterization results it emerges that, as detailed in Section 4, the developed calibration system can be used to test voltage transformers with an accuracy up to 0.1. On the contrary, when only nominal accuracy specifications are available, the accuracy class of the inductive VTs under test is limited to 0.5 or worse. This means that even using the same setup, the adopted approach affects the uncertainty of the results. Finally, the influence of the random effects acting on the measurement setup has been evaluated and compared to the overall accuracy. As a final comment, what was obtained for voltage transformer can also be replicated for current transformers. However, the obtained results cannot be directly reflected to them because they need to be tested with quite a different measurement setup.

## Figures and Tables

**Figure 1 sensors-19-04472-f001:**
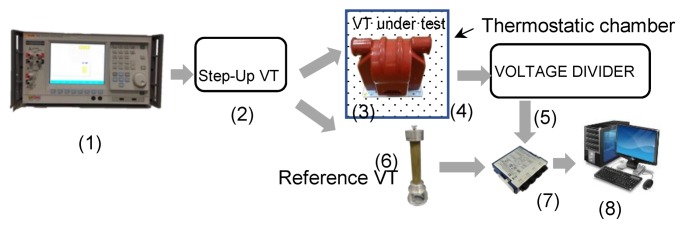
Experimental setup, adapted from [11], used for testing the two voltage transformers(VTs).

**Figure 2 sensors-19-04472-f002:**
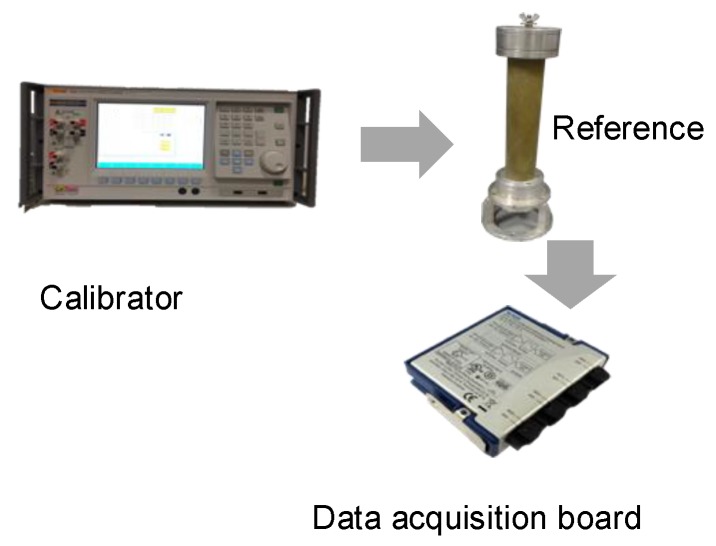
Simple schematic of the setup adopted for the reference divider characterization.

**Figure 3 sensors-19-04472-f003:**
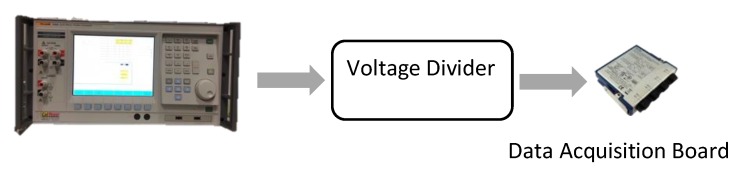
Simple representation of the setup adopted to characterise the resistive divider.

**Figure 4 sensors-19-04472-f004:**
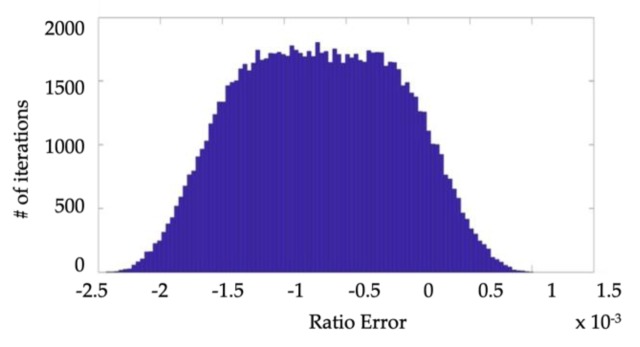
PDF of the ratio error calculated from the nominal accuracy specifications of the items.

**Figure 5 sensors-19-04472-f005:**
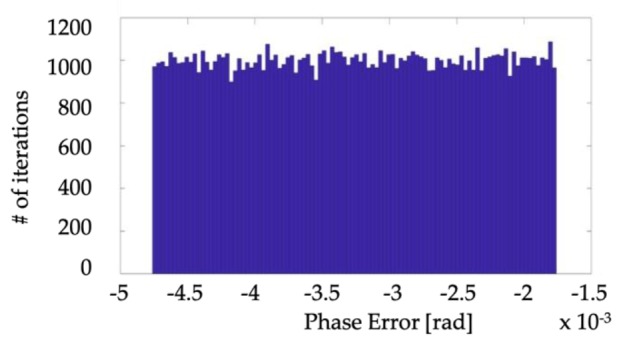
PDF of Δ*φ* calculated from the nominal accuracy specifications of the items.

**Figure 6 sensors-19-04472-f006:**
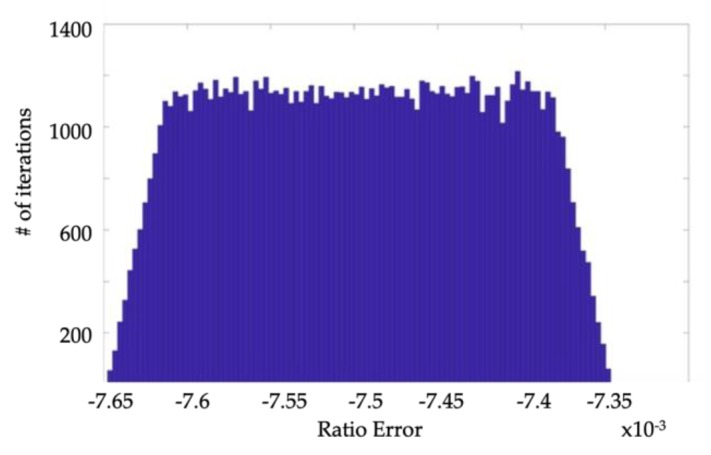
PDF of the *ε* calculated from the metrological characterization of the items.

**Figure 7 sensors-19-04472-f007:**
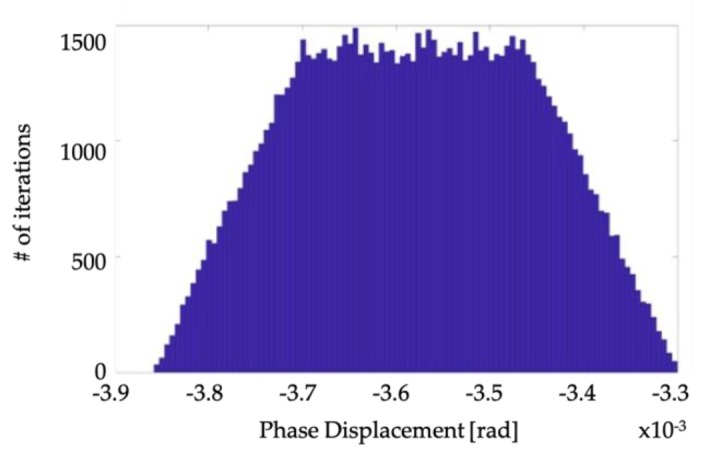
PDF of the Δ*φ* calculated from the metrological characterization of the items.

**Figure 8 sensors-19-04472-f008:**
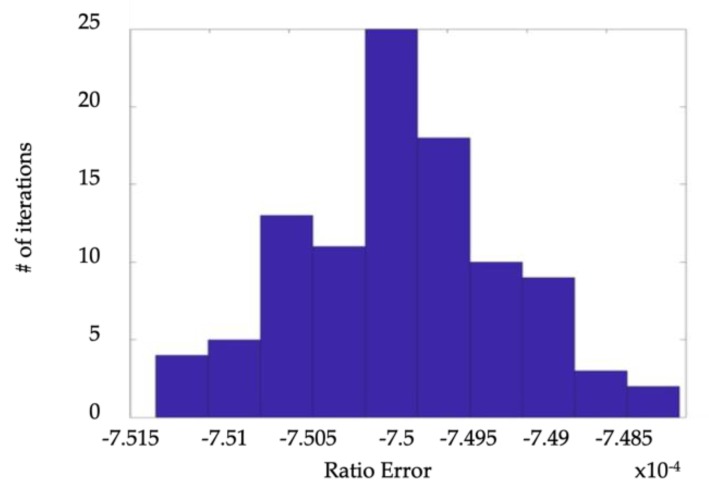
PDF of the measured Phase Error obtained in [11] for the case recalled in Table 4.

**Table 1 sensors-19-04472-t001:** Nominal accuracy specifications of the items which compose the calibration setup.

	R-C Divider	R Divider	DAQ (Data AcQuisition Board)
Nominal Ratio	$K_{r}=$ 5981	$K_{d}=$ 11	-
Phase Error (mrad)	1.5	0	-
Ratio Error	0.1%	0.02%	-
Input Noise (µV)	-	-	70
Gain Error	-	-	0.03%

**Table 2 sensors-19-04472-t002:** Actual ratio and phase displacement of the R-C divider + DAQ measurement chain, together with their standard deviations when 1000 V are applied.

Day	${\tilde{k}}_{RC}$ [-]	$\sigma_{kRC}$ [-]	${\tilde{\varphi }}_{RC}$ [mrad]	$\sigma_{\varphi RC}$ [mrad]
*1*	5984.31	0.08	0.19	0.09
*2*	5983.61	0.07	0.21	0.07
*3*	5983.47	0.07	0.20	0.08
*4*	5983.19	0.08	0.25	0.07
*5*	5981.74	0.06	0.23	0.02

**Table 3 sensors-19-04472-t003:** Actual ratio and phase displacement of the R-divider + DAQ measurement chain, together with their standard deviations when 57 V are applied.

Day	${\tilde{k}}_{R}$ [-]	$\sigma_{kR}$ [-]	${\tilde{\varphi }}_{R}$ [mrad]	$\sigma_{\varphi R}$ [mrad]
*1*	11.0022	1 $\times 10^{-4}$	−0.1	0.2
*2*	11.0024	2 $\times 10^{-4}$	−0.1	0.2
*3*	11.0025	1 $\times 10^{-4}$	0.0	0.1
*4*	11.0023	2 $\times 10^{-4}$	0.0	0.1
*5*	11.0025	1 $\times 10^{-4}$	−0.1	0.2

**Table 4 sensors-19-04472-t004:** Measured ratio and phase error obtained in [11] and used as a reference for this study.

TUT (Transformer under Test)	Temperature [°C]	Quantity	Mean Value	Standard Deviation
*A*	**5**	*ε* [-]	$-7.448\times 10^{-4}$	$6\times 10^{-7}$
		Δ*φ* [mrad]	−3.3	0.7
	**25**	*ε* [-]	$-7.497\times 10^{-4}$	$7\times 10^{-7}$
		Δ*φ* [mrad]	−3.259	$6\times 10^{-3}$
	**45**	*ε* [-]	$-7.837\times 10^{-4}$	$6\times 10^{-7}$
		Δ*φ* [mrad]	−3	1
*B*	**5**	*ε* [-]	$-1.946\times 10^{-3}$	$8\times 10^{-7}$
		Δ*φ* [mrad]	−2.2	0.7
	**25**	*ε* [-]	$-1.971\times 10^{-3}$	$6\times 10^{-7}$
		Δ*φ* [mrad]	−2.143	$6\times 10^{-3}$
	**45**	*ε* [-]	$-1.988\times 10^{-3}$	$3\times 10^{-6}$
		Δ*φ* [mrad]	−2	1

**Table 5 sensors-19-04472-t005:** Results of the Monte Carlo (MC) method application when considering the nominal accuracy specifications for the uncertainty evaluation.

TUT	Temp. [°C]	Quantity	Mean Value	Standard Deviation	Limits of the 95%-Confidence Interval	95%-Confidence Interval Uncertainty
*A*	**5**	*ε* [-]	$-7\times 10^{-4}$	$6\times 10^{-4}$	$-1.9\times 10^{-3}$, $-0.4\times 10^{-3}$	$7.5\times 10^{-4}$
		Δ*φ* [mrad]	−3.3	0.9	−4.7, −1.8	1.45
	**25**	*ε* [-]	$-7\times 10^{-4}$	$6\times 10^{-4}$	$-1.9\times 10^{-3}$, $-0.4\times 10^{-3}$	$7.5\times 10^{-4}$
		Δ*φ* [mrad]	−3.3	0.9	−4.7, −1.8	1.45
	**45**	*ε* [-]	$-8\times 10^{-4}$	$6\times 10^{-4}$	$-1.9\times 10^{-3}$, $-0.4\times 10^{-3}$	$7.5\times 10^{-4}$
		Δ*φ* [mrad]	−3.2	0.9	−4.7, −1.8	1.45
*B*	**5**	*ε* [-]	$-2\times 10^{-4}$	$6\times 10^{-4}$	$-1.3\times 10^{-3}$, $0.7\times 10^{-3}$	$1\times 10^{-3}$
		Δ*φ* [mrad]	−2.2	0.9	−3.6, −0.9	1.35
	**25**	*ε* [-]	$-2\times 10^{-4}$	$6\times 10^{-4}$	$-1.3\times 10^{-3}$, 0.7 × 10^−3^	$1\times 10^{-3}$
		Δ*φ* [mrad]	−2.1	0.9	−3.6, −0.9	1.35
	**45**	*ε* [-]	$-2\times 10^{-4}$	$6\times 10^{-4}$	$-1.3\times 10^{-3}$, $0.7\times 10^{-3}$	$1\times 10^{-3}$
		Δ*φ* [mrad]	−2.1	0.9	−3.6, −0.9	1.35

**Table 6 sensors-19-04472-t006:** Results of the MC method application when considering the metrological characterization of the items for the uncertainty evaluation.

TUT	Temp. [°C]	Quantity	Mean Value	Standard Deviation	Limits of the 95%-Confidence Interval	95%-Confidence Interval Uncertainty
*A*	**5**	*ε* [-]	$-7.4\times 10^{-4}$	$8\times 10^{-5}$	$-7.58\times 10^{-3}$, $-7.32\times 10^{-3}$	$1.29\times 10^{-4}$
		Δ*φ* [mrad]	−3.6	0.1	−3.8, −3.4	$2.23\times 10^{-4}$
	**5**	*ε* [-]	$-7.5\times 10^{-4}$	$8\times 10^{-5}$	$-7.63\times 10^{-3}$, $-7.37\times 10^{-3}$	$1.29\times 10^{-4}$
		Δ*φ* [mrad]	−3.6	0.1	−3.8, −3.4	$2.23\times 10^{-4}$
	**45**	*ε* [-]	$-7.8\times 10^{-4}$	$8\times 10^{-5}$	$-7.96\times 10^{-3}$, $-7.71\times 10^{-3}$	$1.29\times 10^{-4}$
		Δ*φ* [mrad]	−3.5	0.1	−3.8, −3.3	$2.23\times 10^{-4}$
*B*	**5**	*ε* [-]	$-1.95\times 10^{-3}$	$8\times 10^{-5}$	$-2.07\times 10^{-3}$, $-1.82\times 10^{-3}$	$1.30\times 10^{-4}$
		Δ*φ* [mrad]	−2.5	0.1	−2.7, −2.3	$2.24\times 10^{-4}$
	**25**	*ε* [-]	$-1.97\times 10^{-3}$	$8\times 10^{-5}$	$-2.10\times 10^{-3}$, $-1.84\times 10^{-3}$	$1.30\times 10^{-4}$
		Δ*φ* [mrad]	−2.5	0.1	−2.7, −2.2	$2.23\times 10^{-4}$
	**45**	*ε* [-]	$-1.99\times 10^{-3}$	$8\times 10^{-5}$	$-2.12\times 10^{-3}$, $-1.85\times 10^{-3}$	$1.29\times 10^{-4}$
		Δ*φ* [mrad]	−2.4	0.1	−2.6, −2.2	$2.23\times 10^{-4}$

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
