# Peer review of "Uncertainty Analysis of a Test Bed for Calibrating Voltage Transformers Vs. Temperature†"

_sensors, 2019, doi:10.3390/s19204472_

Round 1

Reviewer 1 Report

1) A general measurement System structure is shown in figure 1, but the authors (s) are silent about the proposed measurement System.
figure 1 is very simple. The figure should match the text.

2) It is necessary to describe model with higher quality.

3) It is necessary to increase the conclusion. The conclusion is very small.

4) Newer references are necessary. Your references old. There are many papers in 2012-2019.

Author Response

1) A general measurement System structure is shown in figure 1, but the authors (s) are silent about the proposed measurement System. Figure 1 is very simple. The figure should match the text.

We thank the reviewer for the comment. Each element of the measurement system shown in Fig. 1 is detailed in chapter “2. Measurement System”. To improve readability of the measurement system description, each element in Fig. 1 has been marked with a number and referred in the text accordingly. Moreover, an explanation of the operation of the system has been added in the same chapter.

2) It is necessary to describe model with higher quality.

We appreciate the reviewer’s comment. For the sake of brevity, no details on the models of the measurement processes on which MC method relies on were previously provided. However, now, for each of the developed study (Sections 4.1 and 4.2) full details on the applied methodology have been added (in both sections).

3) It is necessary to increase the conclusion. The conclusion is very small.

The conclusion has been improved to better fit the work’s aim.

4) Newer references are necessary. Your references old. There are many papers in 2012-2019.

New references have been included according to the reviewer’s suggestion.

Reviewer 2 Report

The measurement system and test method in this paper has been published in the proceedings of 2017 IEEE Workshop on AMPS [ref. 9] and a similar work has also been presented in another article of this year "Effect of Temperature on the Accuracy of Inductive Current Transformers". It is difficult to find any new contribution in this paper expect the experiment data and PDF figures. I suggest that the authors should summarize the data again and try to draw some new conclusions. In addition, the abstract need to be rewrote and improved, to avoid including the same conclusion as in [ref.9].

Author Response

The measurement system and test method in this paper has been published in the proceedings of 2017 IEEE Workshop on AMPS [ref. 9] and a similar work has also been presented in another article of this year "Effect of Temperature on the Accuracy of Inductive Current Transformers". It is difficult to find any new contribution in this paper expect the experiment data and PDF figures. I suggest that the authors should summarize the data again and try to draw some new conclusions. In addition, the abstract need to be rewrote and improved, to avoid including the same conclusion as in [ref.9].

According to the reviewer’s suggestion, we improved the overall text to better highlight that:

In [11], the measurement setup used in this work has been studied component by component. In particular, each device has been singularly characterized to evaluate its performance. After that, two voltage transformers have been tested vs. ambient temperature varying in a predefined range. This, as main objective of the work, wanted to highlight a lack in the VTs standards: the accuracy evaluation vs. ambient temperature. In [12] the aim was to evaluate the uncertainty of the measurement setup proposed in [11] by exploiting the characterization of the measurement chain. This, to assess the goodness of the presented setup from a metrological point of view. In [13], mentioned by the reviewer, we studied the effects of ambient temperature on the accuracy parameters of the current transformers (ratio error and phase displacement). Such work used a completely different setup, compared to the one used in the submitted work, and did not tackle the uncertainty analysis of the setup. While this paper deals with the comparison of two methods to evaluate uncertainty on a measurement setup. The first uses the results of the metrological characterization of the elements composing the setup. The second instead, uses as input the accuracy specifications provided by the devices’ manufacturers. In addition, the effects of random effects on the overall uncertainty have been quantified and evaluated.

Reviewer 3 Report

Dear authors,

here are some question and marks which need to be solve. 

Figure 1. Resolution of calibrator, VT, ref VT and NI DAQ card need to be improved. Figure can be the same as presented in previously work if it is referenced. This one has a different calibrator. 

Author’s mentioned in line 94 the temperature-controlled chamber. Is this chamber also used for reference divider characterization presented in Table 3? If not, why not? You sad that some influence quantities have minor daily variation (i.e. room temperature), why didn’t you eliminate that influence with temp-controlled chamber? Same question for R-divider and DAQ measurement setup for characterization of resistive divider.

(line 148) Mean value and st_dev from whole set of 5000 measurements are 5983 and 0,07, all five sets of measurement (presented in table 3) are independent: why didn’t you use weighted arithmetic mean (the result will then be mean 5983,11 with st_dev 0,03)? Same question for mean value and st_dev for R-divider and DAQ presented in Table 4.

Figure 3. Poor figure resolution.

Figure 4 and 5 can be grouped or enlarged separately

Figure 6 and 7 can be grouped or enlarged separately

Figure 8. Can be enlarged

All tables should be uniform in shape, currently authors used three different types of tables.

(line 172) Equation (3) missing comma after expression.

(Line 190) reference [19] is in Italian, can you add (without removing [19]), one additional reference in English so the readers can trace the equation (5) and your uncertainty analysis.

(line 195) Please explain the parameters kx and kc more in detail.

Thank you!

Author Response

Dear authors,

here are some question and marks which need to be solve. 

Figure 1. Resolution of calibrator, VT, ref VT and NI DAQ card need to be improved. Figure can be the same as presented in previously work if it is referenced. This one has a different calibrator. 

Figure 3. Poor figure resolution.

Figure 4 and 5 can be grouped or enlarged separately

Figure 6 and 7 can be grouped or enlarged separately

Figure 8. Can be enlarged

All tables should be uniform in shape, currently authors used three different types of tables.

We thank the reviewer for highlighting these points, all pictures and tables have been improved to increase their quality and readability.

Author’s mentioned in line 94 the temperature-controlled chamber. Is this chamber also used for reference divider characterization presented in Table 3? If not, why not? You sad that some influence quantities have minor daily variation (i.e. room temperature), why didn’t you eliminate that influence with temp-controlled chamber? Same question for R-divider and DAQ measurement setup for characterization of resistive divider.

The thermostatic chamber has been used only for the voltage transformer under test. All other devices, including references, acquisition system, power source, etc. have been appositely kept outside the chamber because the study wanted to focus only on the VT analysis. Therefore, in this first step, we wanted to separate the temperature influence on the overall equipment from the one on the VT. This is why in the text we refer to minor daily changes (0.004 %). As a matter of fact, the room temperature of the laboratory may oscillate between 22-24 degrees, but without significant consequences on the performed measurements.

(line 148) Mean value and st_dev from whole set of 5000 measurements are 5983 and 0,07, all five sets of measurement (presented in table 3) are independent: why didn’t you use weighted arithmetic mean (the result will then be mean 5983,11 with st_dev 0,03)? Same question for mean value and st_dev for R-divider and DAQ presented in Table 4.

We thank the reviewer for the comment. In the cited tables, data come from 5 identical set of measurements performed in 5 consecutives days. Therefore, we believe that the performed arithmetic mean is the best way to approach the obtained results. In other words, no significant reasons have been found to give more weight to a day compared to another one. This, considering that nothing has been intentionally changed from one measurement to the following one.

 (line 172) Equation (3) missing comma after expression.

Corrected, thank you.

(Line 190) reference [19] is in Italian, can you add (without removing [19]), one additional reference in English so the readers can trace the equation (5) and your uncertainty analysis.

According to the reviewer’s request we added a new reference for (5).

(line 195) Please explain the parameters kx and kc more in detail.

In the text a further explanation of the parameters has been added:

“In other words,  and , represent the known ratios of the devices used to perform the measurement of  and  (necessary for the ∆ε computation).”

Round 2

Reviewer 2 Report

The authors have improved the paper in abstract, content and conclusion. The revised version gives a more clear description about the research work and its background, which meets the requirement for publication.